# Relationship between Species Richness, Biomass and Structure of Vegetation and Mycobiota along an Altitudinal Transect in the Polar Urals

**DOI:** 10.3390/jof6040353

**Published:** 2020-12-09

**Authors:** Anton G. Shiryaev, Ursula Peintner, Vladimir V. Elsakov, Svetlana Yu. Sokovnina, Denis A. Kosolapov, Olga S. Shiryaeva, Nadezhda M. Devi, Andrei A. Grigoriev

**Affiliations:** 1Vegetation and Mycobiota Biodiversity Department, Institute of Plant and Animal Ecology, Ural Branch of the Russian Academy of Sciences, 8 March Str., 202, 620144 Ekaterinburg, Russia; olga.s.shiryaeva@gmail.com; 2Institute of Microbiology, Innsbruck University, Technikerstr. 25, 6020 Innsbruck, Austria; ursula.peintner@uibk.ac.at; 3Komi Scientific Centre, Northern Flora and Vegetation Department, Institute of Biology, Ural Branch of the Russian Academy of Sciences, Kommunisticheskaya Str., 28, 167982 Syktyvkar, Russia; elsakov@ib.komisc.ru (V.V.E.); kosolapov@ib.komisc.ru (D.A.K.); 4Arctic Research Station, Institute of Plant and Animal Ecology, Ural Branch of the Russian Academy of Sciences, Zelenaya Gorka Str., 21, 629400 Labytnangi, Russia; sokovnina_su@ipae.uran.ru; 5Dendrochronology Department, Institute of Plant and Animal Ecology, Ural Branch of the Russian Academy of Sciences, 8 March Str., 202, 620144 Ekaterinburg, Russia; nadya@ipae.uran.ru; 6Geoinformation Technologies Department, Institute of Plant and Animal Ecology, Ural Branch of the Russian Academy of Sciences, 8 March Str., 202, 620144 Ekaterinburg, Russia; grigoriev.a.a@ipae.uran.ru

**Keywords:** biodiversity, fungal ecology, climatic gradient, productivity, flora, lichen, mosses, life form, phytocoenology, plant–fungal interactions, timberline, tundra, Arctic greening

## Abstract

Aboveground species richness patterns of vascular plants, aphyllophoroid macrofungi, bryophytes and lichens were compared along an altitudinal gradient (80–310 m a.s.l.) on the Slantsevaya mountain at the eastern macroslope of the Polar Urals (Russia). Five altitudinal levels were included in the study: (1) Northern boreal forest with larch-spruce in the Sob’ river valley habitats; (2–3) two levels of closed, northern boreal, larch-dominated forests on the slopes; (4) crook-stemmed forest; (5) tundra habitats above the timberline. Vascular plant or bryophyte species richness was not affected by altitudinal levels, but lichen species richness significantly increased from the river valley to the tundra. For aphyllophoroid macrofungi, species richness was highest at intermediate and low altitudes, and poorest in the tundra. These results indicate a positive ecotone effect on aphyllophoroid fungal species richness. The species richness of aphyllophoroid fungi as a whole was neither correlated to mortmass stocks, nor to species richness of vascular plants, but individual ecological or morphological groups depended on these parameters. Poroid fungal species richness was positively correlated to tree age, wood biomass and crown density, and therefore peaked in the middle of the slope and at the foot of the mountain. In contrast, clavarioid fungal species richness was negatively related to woody bio- and mortmass, and therefore peaked in the tundra. This altitudinal level was characterized by high biomass proportions of lichens and mosses, and by high litter mortmass. The proportion of corticoid fungi increased with altitude, reaching its maximum at the timberline. Results from the different methods used in this work were concordant, and showed significant patterns. Tundra communities differ significantly from the forest communities, as is also confirmed by nonmetric multidimensional scaling (NMDS) analyses based on the spectrum of morphological and ecological groups of aphyllophoroid fungi.

## 1. Introduction

The consequences of global climate change are most obvious in the Arctic, where warming rates are two to three times higher than the world average [1]. Vegetational changes are quite evident at these high-latitudes, and get clearly manifested in the range expansion of typical boreal species to the north, and in a reduction in Arctic-Alpine species [2,3]. Such a greening of vegetation takes place throughout the Arctic [4,5]. Concomitantly, major changes were reported for the diversity and composition of animals and vascular plants, but little attention has been paid to fungi up to now [6,7].

Climate change has had a significant impact on the mycobiota at high latitudes over the last 20 years. Long-term fruitbody monitoring studies proved that with warming, species richness and yield of edible fungi increased in Europe, together with a prolongation of the fruiting season [7]. A migration of southern fungal species to the North was reported, with migration rates of about 500–700 km in 10–20 years [8]. Climate change is also affecting subarctic areas in Russia. As shown by our recently published study carried out in the Polar Urals [9], the climate in the region has warmed by 2 °C over the last 60 years. As a consequence, the species richness of aphyllophoroid fungi detected in the Slantsevaya mountain area nearly doubled (from 157 to 257 species). At the same time mushrooms appeared, whose appearance was obviously connected with the emergence of “new boreal” substrates, which were not present before on the mountain, since the vegetation 60 years ago was forest tundra. So the permafrost began to thaw 2 times deeper. At the same time the plant productivity increased enormously: 21 times more woody biomass was produced, which resulted in a sufficient amount of large-sized dead wood (and not only many small spruces and larches). Furthermore, a distinct cover of mesophilic tall grass developed, which overgrew the mossy plant communities of the moor tundra.

The newly recorded fungal species mostly represented typical boreal morphotypes (poroid) and boreal ecological-trophic groups (mycorrhizal fungi, humus-inhabiting saprobes, wood-inhabiting saprobes, etc.). Hand in hand with this increase in boreal species, the proportion of arctic-alpine litter-inhabiting species decreased significantly.

Surprisingly, for a few places in the Polar Urals, it was reported that neither the vegetation, biomass stock nor aphyllophoroid fungal species richness and their ecomorphological structure were affected by climate change [10,11]. This clearly shows that the emergence of new (boreal) fungal species is primarily associated with a change in the vegetation structure and the amount of available plant biomass (biotic factors), and not to the increase in temperature (abiotic factor). The influence of climate on the diversity of aphyllophoroid fungi of the Polar Urals was presented in our previous article [9]. In this context, we are now addressing key factors of the vegetation structure and biomass stock, which are shaping subarctic fungal communities. Based on the previous study [9], changes in the above-ground mycobiota occur in the sequential order: first, temperatures and precipitation increase. In response, the permafrost thaws deeper and the microbial soil activity increases. The vegetation also changes at this stage: its greening, the borealization, takes place. With increased plant productivity, the plant biomass and the especially woody mortmass also increases. The structure of the aboveground mycobiota changes only from this time on.

The Subarctic is one of the simplest organized ecosystems of the planet. Thus, it can serve as a model ecosystem for assessing the influence of basic biotic factors on aphyllophoroid fungal communities. The following globally relevant hypotheses (H) can be addressed in a straight-forward way: H1: Species richness of aphyllophoroid fungi is positively correlated to substrate availability. Thus, the richest mycobiota develops in the floristic richest regions [11,12]. H2: Fungal and vascular plant species richness decreases with altitude in the Polar Urals [13]. H3: The species richness of heterotrophic fungi is positively correlated to the volume of mortmass [14,15]; the higher the mortmass of trees, the higher the species richness of wood-destroying fungi, and the higher the mortmass of the litter, the higher the species richness of litter saprobes. H4: Fungal species richness, especially richness of wood-degrading fungi, is positively related to forest age and crown density [16,17]. H5: Fungal species forming large fruitbodies are associated either to bigger large-sized woody substrates (wood-degrading fungi), or to deeper soil thawing (mycorrhizal fungi) [18,19].

The aim of this study was to address the biotic factors influencing the aboveground species richness and the ecological-morphological structure of aphyllophoroid fungi communities along an altitudinal transect of the Polar Urals. These data should serve as a solid base for modeling future developments in the Arctic. Accurate prediction should not only include changes in vegetation (representing the productive part), but also changes in aphyllophoroid fungal communities (representing the destructive part).

## 2. Materials and Methods

### 2.1. Study Area

The study area is located where the Sob’ river cuts through the eastern slope of the Polar Urals at the territory of the Yamal-Nenetsk Autonomous District in Russia. The river flows from northwest to southeast, with the valley framed from west and southwest by Rai-Iz mountain, and from north and northeast by the Slantsevaya mountain. The area is located 30 km north of the Arctic Circle (N 66°54′; E 65°44′) and the border between Europe and Asia, 60 km east of Vorkuta town (Komi Republic), 50 km west of Salekhard town (capital of Yamal-Nenetsk Autonomous District) and 30 km west of Labytnangi town (Figure 1). The study site has an area of 10 km^2^ (the width of the river valley with mountain slopes is 2 km and 5 km length along the river).

The digital relief model of the study area was taken from ArcticDEM (http://www.pgc.umn.edu/data/arcticdem/). This model is used for topographic correction of figures of crown density, relief and types of vegetation.

Monthly meteorological records covering the period from 1892 to 2018 are available from the nearest weather station of Salekhard (N 66°32′, E 66°32′; 35 m a.s.l.). These data indicate a negative average annual temperature (−6.4 °C) and a mean annual precipitation of 415 mm, of which 45% falls as snow (Figure 2). The region is characterized by a complex wind regime dominated by westerly winds, with an average wind speed of 8.5–8.8 m/s in winter, and 6.5–7.0 m/s in summer.

There has been almost no economic activity in the study area over the last 60 years. A small village (14 houses) was located here from 1940–1950, where the construction workers of the Salekhard–Vorkuta railway lived, which currently crosses the Polar Urals from east to west. At the moment, anthropogenic activity here is restricted to transhumance by the local Nenets driving herds of reindeer during winter. Under the Slantsevaya mountain there are two houses (rail station 141 km) in which two people permanently live [9].

The response of the vegetation to climate warming is well studied in the Polar Urals [2,9,20,21,22]. Over the 60 years that have passed since the beginning of these long-term monitoring studies, the climate has changed significantly in the Slantsevaya mountain are: the average annual summer temperature has increased by 1.8 °C, and the winter temperature by 2.0 °C, and, as a consequence, the length of the growing season has increased by 7 days. The biomass of the dominating tree species has increased significantly. For *Picea obovata* Lebed. from 0.22 to 4.64 t/ha, and for *Larix sibirica* Lebed. from 28.34 to 97.60 t/ha. The crown density of the trees has also increased by 20% [9,22]. Photographs taken in 1962 and in 2020 on the slopes of Slantsevaya mountain show a drastic transformation of the vegetation in the river valley and on the mountain slopes from forest-tundra to north boreal forest [2], and impressively document the rising of the upper forest boundary by 50 m over 58 years (Figure 3). Particularly noteworthy is the afforestation of the gentle slope in the front part of the image, as it is clearly associated with an increase in density and productivity of these larch stands. A similar trend was noted in the neighboring mountains [2,20].

### 2.2. Sampling Sites and Altitudinal Gradient

Plant communities were studied on an altitudinal gradient (transect) on the western/ southwestern slope of the Slantsevaya mountain (Figure 4), ranging from the Sob’ river valley to the mountain tundra (80–310 m a.s.l.). The altitudinal gradient included 5 altitude levels: level I (80 m a.s.l.) closed the north boreal forest in the Sob’ river valley; level II (170 m a.s.l.) the middle part of the slope with the closed north boreal forest; level III (230 m a.s.l.), the upper limit of the closed forest; level IV (260 m a.s.l.) the mountain crooked forest; level V (310 m a.s.l.) mountain bush tundra.

Within each altitudinal level, five plots of 400 m^2^ (20 m × 20 m) were established for our phytocoenological investigations. Thus, the sum plots an area of 2000 m^2^ for each level (Appendix A). Within each altitudinal level, the difference in altitude of the plots was plus/minus 8 m. On average, there was a distance of 20 m between each plot. Areas were not included in the study if they had open zones with a crown density differing by more than 20% from the model plots. This was the case if bogs occurred between forest plots. Thus, each altitudinal level has a total area of 3600 m^2^ (20 m × 180 m). In total, along the entire altitude gradient, studies were carried out on an area of 18,000 m^2^ (1.8 ha). The plots included a complex of the most typical phytocenoses for each altitudinal level.

### 2.3. Species Richness and Biomass for Vascular Plants, Bryophytes and Lichens

Field work was carried out in August 2020. On each trial plot, geobotanical descriptions were carried out according to the standard technique [23]: the species composition of plants for the herb-dwarf shrub and the moss-lichen aboveground layers, and the projective cover and the height of the tiers were taken into account. Vascular plant species were assigned to seven functional groups: tall shrubs (*Duschekia fruticosa* (Rupr.) Pouzar, *Salix phylicifolia* L., et al.), shrubs (*Betula nana* L., *Vaccinium uliginosum* L., et al.), dwarf shrubs (*Atragene sibirica* L., *Dryas octopetala* L., *Vaccinium myrtillus* L., et al.), evergreen ericoid (*Andromeda polifolia* L., *Empetrum hermaphroditum* Hagerup, et al.), forbs (*Aconitum septentrionale* Koelle, *Dryopteris carthusiana* (Vill.) HP Fuchs, *Equisetum arvense* L., *Rubus chamaemorus* L., et al.), grasses (together with sedges—*Calamagrostis* sp., *Carex* sp., *Luzula wahlenbergii* Rupr., *Poa* sp., et al.), ericoid plants (*Lycopodium dubium* Zoega), as well as bryophyts and lichens. A more detailed description of the belonging of herbaceous plant species to a specific functional group can be found in [24]. Trees are excluded from this list, as they were studied separately, in the dendrochronology section. The assignment of vascular plant species to six life forms (phanerophytes, hemicryptophytes, chamaephytes, geophytes, therophytes, cryptophytes) follows [25]. The species richness of bryophytes and lichens was estimated for the most common aboveground species only (epilithic species excluded).

Species were subdivided into separate functional groups and life forms, and then their biomass was assessed. Inside three plots at each altitudinal level, three mini-plots (altogether nine in the level) with a size of 50 cm × 50 cm (0.25 m^2^) were studied. All plants growing at the survey mini-plots were harvested, sorted by groups and dried in laboratory drying cabinets to dry-weight constancy. Moreover, leaf and grass litter, woody litter and fallen branches, conifer cones and tree-bark were collected. Samples were weighed on a Kern 440-33 N balance with an accuracy of 0.1 g. For the herb-dwarf shrub layer, the analysis was carried out individually for different life forms and functional groups of vascular plants and litter. The dry-weight of these samples was combined into total mortmass. The biomass and mortmass were also studied for the all moss-lichen aboveground layer (green mosses, Sphagnums and lichens), and the projective cover was calculated. The projective cover and weight for different functional groups and life forms were calculated as the sum of coverings or masses of individual species. Mean and standard error are calculated for 5 plots within each elevation.

Meso- and nanophanerophyts also include woody plants of the herbaceous-shrub layer. In this case, macro-phanerophytes (for example, *Larix sibirica*, *Picea obovata*, *Betula pubescens*, etc.) were separately considered as an element of the first tree layer. The methodology for studying the structure of the tree layer and the dendrochronological work are described below.

### 2.4. Data Sampling and Processing for Dendrochronological Studies

The structure of the tree layer and its biomass and mortmass were studied at all five altitudinal levels (Figure 4). At each level, inside the same 5 plots, were tested 3–5 plots with a size of 20 m × 20 m, on which the structure of the grass-dwarf shrub and moss-lichen layers was also assessed. At level II, the dendrochronological study was carried out in part, while we present data on biomass only.

The morphometric parameters were measured for a total of 694 trees and undergrowth units, and the plant age was calculated for each of them. The trees on the lower levels (I and II) were studied in 2020, but the data from the upper levels (III-V) were taken from our previous studies [9,22].

On each plot, the exact locations of each tree and understory plant were recorded, as well as their height, basal diameter, diameter at an altitude 1.3 m, crown diameter in two mutually perpendicular directions using a tape measure (0.5 m accuracy), a telescopic ruler (1.0 m accuracy) and digital rangefinder (accuracy 0.3 m). Tree ages were determined based on wood samples (cores) taken at the stem base using a Haglof borer, or by taking a transect with a handsaw (less than 3 cm in diameter). Each wood sample was glued to a wooden strip, cleaned with a shaving blade, and pigmented with white powder to enhance ring boundary contrasts [26]. Growth ring counting and core dating were carried out according to generally accepted methods [26,27] under laboratory conditions. All wood samples were measured on a semi-automatic machine Lintab 6 (F. Rinn S.A., Heidelberg, Germany) [28]. To identify false and missing rings, wood samples were dated according to the generalized tree-ring chronology for the study site [3,29]. Trees less than 1.5 m and of an age less than 30 years were attributed to saplings.

If the collected drill holes did not reach the center of the trunk, the age of the tree was extrapolated based on local tree ring patterns. The ratio of tree age to height was determined using a regression equation. Using this data, corrections could be calculated to determine the more accurate age of each of the trees we examined with a diameter >3–4 cm [26]. We made sure that no significant number of dead trees or stump remains remained at the site after the trees were felled. We could not find any signs of fire in our drill cores. From this, we conclude that no forest fires have occurred on the test plots in the last 400 years. Other unfavorable factors can also be excluded due to the structure of the annual rings.

The aboveground tree biomass was estimated for the two most widespread tree species in the study area: Siberian larch (*Larix sibirica*) and common spruce (*Picea obovata*) according to methods described before [21,22,29]. For the upper forest boundary and the upper enclosed forest, the tree biomass estimates were conducted at the border of the trial-plots. The so-called model trees (*n* = 33) were felled and sectioned. The stem biomass was determined with hand scales with accuracy 50 g. The fresh weight of leafless branches, needles and generative organs were determined separately with a digital weight (accuracy 0.01 g). All subsamples were then oven-dried at 106 °C to stable weight. The drying time ranged from several hours to days. Then, the masses of all these biomass fractions were summarized. Allometric functions between the total tree biomass (dry-weight) and the tree diameters at base height were calculated [22]. Using these data together with the tree ages, the annual biomass accumulation could be calculated for each tree. The calculations were based on the following formula:D*_n_* = (R*_n_*/R*_final_*) × D*_final_*

D*_n_* is the computed tree diameter in a certain year; R*_n_* is distance from the tree core to a certain annual ring; R*_final_* is the current tree radius; and D*_final_* is the current tree diameter. Now, based on this database, the aboveground biomass of larch and spruce per hectare could be reliably determined based on the measurements carried out on the trees growing in our plots. In addition, the biomass of birch trees (*Betula pendula*) was determined based on previously published relationships and equations [21] for the plot levels I and II. The significance of birch for the structure of the stand increased significantly at lower levels, but at upper levels, large-stemmed birch trees occurred sporadically or were absent.

The mortmass parameters were studied for coarse woody debris (CWD, diameter more than 5 cm: dead trees) and fine woody debris (FWD, less than 5 cm: branches, thin trees, cones and bark pieces) at all altitudinal levels in 2020. Mortmass was determined based on measurements of dead wood volume and its density for each species, separately. For CWD, the mortal mass was calculated for dead fallen and standing trees. On the high-altitude profile (V), dead trees were found only scattered in the mountain tundra, and there were only branches of shrubs (*Betula nana*, *Salix* spp.), while at the foot of the mountain, dead trees prevailed over shrubs.

### 2.5. Fungal Sampling

Aphyllophoroid fungi were used in this study as a model group for macromycetes. The first samples in the study area were collected by S.G. Shiyatov and N.T. Kazantseva in 1959, but regular research began in 1961. The results of 60 years of mycological monitoring in the Sob’ river valley and on the slopes of the Slantsevaya mountain were recently published [9]. These baseline data were now complemented by our most recent data from August 2020, when we carried out extensive sampling along the altitudinal gradient established in this study. A list of aphyllophoroid fungi and the distribution of species by altitudinal levels is given in Appendix A. Furthermore, the list of species includes information concerning fruitbody types, ecological strategies and height groups.

Fruitbody types, or morphotypes, of aphyllophoroid fungi include three major species-rich groups (corticioids, poroids and clavarioids), and some smaller groups (e.g., cantharelloids, thelephoroids and stipitate hericioids). These groups are good indicators of bioclimatic conditions: in the tundra without autochthonous woody substrates, clavarioids are dominant; in shrub tundra with many different small twigs, corticioids on *Betula nana* bushes as well as a lot of clavarioids on different grasses and herbs are dominating. In the boreal forests with many big coniferous and deciduous tree trunks, poroids have a big share of the aphyllophoroid species richness.

Ecological strategies of aphyllophoroid fungi include three basic types. Parasites (on alive trees, shrubs, grasses and mosses), mutualistic symbionts (ectomycorrhiza-forming and basidio-lichens) and saprobs (on deadwood, humus and litter). Fungi growing on dead grass, leaves, needles, mixed decaying litter, twigs less than 1 cm in diameter, as well as on wood in its final decay stage 5 (representing individual fibers) or on a separate bark (for example, inside the birch bark) were defined as litter saprobs. Among the saprobes, we did not distinguish between species with substrate types on mosses, as there were only two species and there were no dynamics. Despite the fact that parasites develop on different substrates, we did not subdivide this group into separate substrate groups, because almost all species were collected on wood, and only one species on grasses. We also considered symbionts as a single group, since there are were only two species of basidiolichens, and they were found only in the mountain tundra.

A few fungi are mixotroph (e.g., genus *Tomentella*—ectomycorrhizal and litter/wood saprobes, *Osteina obducta* (Berk.) Donk—wood and soil saprobs, etc.) In this case, they were listed in both groups. In order to obtain generalized data, the respective shares were normalized to a common denominator (100%). In general, the ratio of these three ecological groups varies significantly depending on the latitudinal-zonal position of the studied region [9].

Height groups, or fruitbody size, were estimated only for species with negatively geotropic fruit bodies >2 mm (clavarioids, cantharelloides, thelephoroids and stipitate hydnoids). They are divided into three groups: group I—fruit bodies up to 3 cm high, group II—up to 8 cm and group III—more than 8 cm high. Small-fruiting species (group I) prevail in the arctic tundra, while in rich hemiboreal and nemoral forests with tall trunk trees, the proportion of large-fruiting species (groupIII) is high [9,10].

The fungi were collected at the same five altitudinal levels, each with an area of 20 by 180 m (Figure 4). Averaging over the plots within these levels was not done. The fruitbodies collected in 2020 were deposited in the following mycological collections: Institute of Plant and Animal Ecology UrB RAS, Ekaterinburg (SVER) and Institute of Biology, Komi SC UrB RAS, Syktyvkar (SYKO).

### 2.6. Data Analysis

Ordination analyses were carried out on non-metric multidimensional scaling (NMDS) in the ExStatR program [30]; the statistical package R (V. 3.5.2) was used for visualization [31]. The significance of differences between groups was calculated based on the nonparametric Kruskal–Wallis test—a comparison of median values. Data were tested for significant differences based on nonparametric Kruskal–Wallis tests, Spearman rank correlation (*r*_s_) and the Monte Carlo method (U).

## 3. Results

### 3.1. Vegetation Cover and Plant Communities

The highest species richness of vascular plants was detected in the river valley (46 species), and the lowest in the tundra (26). In the intermediate altitudinal layers, the number of species varied from 37 to 38 (Table 1).

The altitudinal level I is characterized by a closed north boreal forest in the Sob’ river valley, with *Picea obovata*, *Betula pendula* and *Larix sibirica*. *Duschekia fruticosa* and occasional *Sorbus sibirica* Hedl. shrubs are dominated by *Ribes rubrum* L. and scattered *Betula nana*, *Juniperus communis* L., *Lonicera caerulea* L., *Rosa acicularis* Lindl., *Salix phylicifolia* and *Vaccinium uliginosum*. Among the dwarf shrubs, *Atragene sibirica* L. and *Vaccinium vitis-idea* L. are present. The dominant grasses are *Calamagrostis* sp., *Equisetum arvense*, *Geranium pretense* L. Dominant mosses are *Hylocomium splendens* (Hedw.) Bruchetal and *Dicranum* spp.

In the middle part of the slope (altitudinal level II), communities of closed north boreal forest with *Larix sibirica* and *Picea obovata* are dominating. The understory consists of *Betula pendula*, *Duschekia fruticosa*, *Salix phylicifolia* and *Sorbus sibirica*. The shrub layer is formed by *Linnaea borealis* L., and single *Lonicera caerulea*, *Ribes rubrum*, *Rosa acicularis*. Dominant shrubs are *Andromeda polifolia*, *Vaccinium myrtillus*, and *V. vitis-idea*. The dominant herbs are *Aconitum septentrionale*, *Calamagrostis* sp., *Chrysosplenium sibiricum* (Ser. ex DC.) A.P. Khokhr., and *Dryopteris carthusiana*. Dominant mosses are *Dicranim* spp. and *Hylocomium splendens*, *Polytrichum* spp., *Sanionia uncinata* (Hedw.) Loeske.

In the upper limit of the closed forest (altitudinal level III), *Larix sibirica* dominates, with only single individuals of *Picea obovata*. The undergrowth consists of *Duschekia fruticosa* and *Sorbus sibirica*. The shrub layer is unevenly formed from *Betula nana*, singly *Linnae aborealis*, *Lonicera caerulea*, *Rosa acicularis*, *Salix pyrolifolia*. Shrubs are represented by *Empetrum hermaphroditum*, *Ledum palustre* L., *Vaccinium myrtillus*, *V. vitis-idea*. The dominating herbs are *Aconitum septentrionale*, *Calamagrostis* sp., *Cardamine amara* L. and *Poa* sp. The ground cover is formed by mosses of the genera *Dicranim*, *Polytrichum* and by *Hylocomium splendens*.

The crooked forest belt (altitudinal level IV) consists of larch and spruce, and thickets of *Duschekia fruticosa* and *Salix hastata* L. *Betula nana* and *Salix phylicifolia* are present in the shrub layer. Among the shrubs, the most abundant ones are *Vaccinium myrtillus* and *V. vitis-idea*. In the thickets of *Duschekia fruticosa* and *Salix hastata*, the shrub layer is missing. The dominant grass species at this altitude level are *Aconitum septentrionale*, *Calamagrostis* sp., *Cardamine amara*, *Equisetum arvensis*, *Rubus saxatilis* L. Genus *Dicranim*, *Mnium*, and *Hylocomium splendens* dominate among the mosses.

The mountain-tundra belt (altitudinal level V) is a shrub tundra with grasses, lichens and mosses, as well as a hilly bog. In the communities of the mountain-tundra belt, *Larix sibirica*, *Picea obovata* and *Duschekia fruticosa* occur as single trees. *Betula nana* and *Vaccinium uliginosum*, and scattered *Salix phylicifolia* form the shrub layer. Among the shrubs, *Andromeda polifolia*, *Empetrum hermaphroditum* and *Ledum decumbens* dominate. The dominant grasses are *Bistorta major* Gray., *Calamagrostis* sp., *Rubus chamaemorus*. Dominant mosses are *Hylocomium splendens*, *Ptilidium ciliare*, *Sphagnum* sp. Among the lichens, the most abundant ones are *Cladonia amaurocraea* (Flörke) Schaer, *C. arbuscula* (Wallr.) Flot., *C. rangiferina* (L.) FH Wigg., *C. stygia* (Fr.) Ruoss, *C. uncialis* (L.) Weber ex FH Wigg. The vertical distribution of vegetation on the studied mountain is similar to the neighboring mountain ranges [13].

All investigated plant communities had a well-developed projective ground cover (>80%). The largest proportion of litter and undergrowth (up to 17%) was found in the lower part of the mountain forest belt. Rock outcrops were only found in the mountain tundra belt. Insignificant areas of bare soil (0.7–3.6%) were identified at all levels (Appendix A).

The tundra belt had the smallest cover of grass-dwarf shrub layer (F (4.18) = 9.40; *p* < 0.01); otherwise there were no differences in projective cover between different altitudinal levels. Tundra phytocenoses had a significantly lower projective cover of grass than any other studied level (F (4.18) = 25.01; *p* << 0.01). Here, the largest projective cover of the moss-lichen layer was 90–100% (F (4.18) = 8.55; *p* < 0.01), and only in these communities did lichens contribute to layer formation (PP up to 50%) (F (4.18) = 16.59; *p* < 0.01). Moreover, there is a significantly higher projective cover of shrubs (F (4.18) = 3.88; *p* = 0.02) and evergreen ericoid plants (F (4.14) = 3.25; *p* = 0.04) in the tundra, while the share of forbs (F (4.18) = 9.48; *p* < 0.01) and cereals (F (4.18) = 4.08; *p* = 0.02) is minimal. The maximum cover of forbs was found in the upper part of the mountain forest (F (4.18) = 9.48; *p* < 0.01). In the middle part of the mountain-forest belt, the maximum coverage of grasses (F (4.18) = 0.08; *p* = 0.02) and ericoid plants (F (4.6) = 0.83; *p* = 0.01) occurred. Ground lichens are absent in forest communities. The main parameters of the projective cover of the grass-shrub and moss-lichen cover on the Slantsevaya mountain are similar to those of other studied mountains on the eastern slope of the Polar Urals [32,33].

The distribution of different life forms of plants correlates with the distribution of functional groups. The plant communities of the tundra belt have a high proportion of nanophanerophytes (F (4.13) = 5.63; *p* = 0.01) (Appendix A). The smallest proportion of hemicryptophytes was found in the tundra communities and floodplain forests (F (4.18) = 12.44; *p* < 0.01). Floodplain forests had significantly higher proportions of geophytes than the other altitudinal levels of tundra and the upper and lower levels (F (4.14) = 4.75; *p* = 0.01). Tundra communities have the lowest proportion of therophytes compared floodplain forests (F (4.13) = 1.92; *p* = 0.03), and also have significantly lower proportions of cryptophyte to forests communities at lower levels (F (4.11) = 1.60; *p* = 0.03).

Evergreen ericoid plants form significantly higher phytomass in tundra communities than in forest communities of different altitudes (F (4.8) = 8.40; *p* = 0.01), and the proportion of fallen branches and cones is also significantly lower here (F (4.11) = 11.98; *p* < 0.01). Grasses form the highest biomass in mid-level phytocenoses (F (4.9) = 4.74; *p* = 0.03) (Table 2).

The biomass fraction of meso- and nanophanerophytes is maximum in the middle level of forests, and is minimum in the phytocenoses of the lower levels (F (4.8) = 6.1; *p* = 0.02). The cryptophyte biomass decreases significantly with the transition from the tundra to the upper forest line, and from the upper forest line to other investigated levels of mountain forests (F (4.7) = 27.94; *p* < 0.01). The maximum biomass of hemicryptophytes was found at level II (Appendix A). The distribution of plant biomass along mountain belts is similar to the indicators established for other East-European Subarctic mountain regions [34].

### 3.2. Trees

With the increasing altitude, the tree species richness decreases (macrophanerophyts): six species were found in the river valley, four at timberline, and only two species in the mountain tundra (*Larix sibirica* and *Picea obovata*). The average morphometric and areal characteristics of the tree stands decrease with altitude (Appendix A). For the two most common tree species in the area (*L. sibirica* and *P. obovata*), the average base diameter decreases by a factor of 3, height by a factor of 6, age by a factor of 3 and crown density by a factor of 9.

With the increasing altitude, the proportion of larch by a factor of 2 and the aboveground biomass of forest stands decreases by a factor 58. In the floodplain of the Sob’ river, a significant proportion of birch is present in the stand, making up almost half of all trees (Appendix A). This explains the proportional decrease in the biomass of larch and spruce in this site.

Plant biomass prevails over mortmass at all forest levels of the transect (forests: 3.33–2.67 times higher), but mortmass prevails in the tundra (3.8 times higher). The sum of the plant biomass with the mortmass gives the total aboveground phytomass, which decreases with altitude by 3 times (Table 3). CWD and FWD form more than half of the whole mortmass at lower altitudinal levels, but in the mountain tundra only 2%, whereas litter biomass is absolutely dominating.

The total mortmass of CWD and FWD is similar at all forest levels due to the small number of dead trees. Coarse woody debris (CWD) is significantly lower in the taiga compared to forests. In the study area, CWD decreases by a third from the valley to the upper forest boundary, and in the mountain tundra, there is a sharp drop to almost zero t/ha (Table 3). Fine wood debris (FWD) dropped by almost 2/3, in the tundra even to 0.7 t/ha (due to the presence of dead branches of willow and birch). In the mountain tundra, the FWD was higher than the CWD. The highest litter mass is produced in the tundra (especially hummock bogs), where it is almost three times higher than in the valley forests. The sum of the plant biomass with the mortmass gives the total aboveground phytomass, which decreases with altitude by a factor 3 (Table 3).

A similar trend was reported for zonal plant communities during the transition from northern forests to forest-tundra and subarctic tundras [35]. Here, CWD and FWD accounted for 54% of mortmass at lower altitudinal levels, but for only 2% in the mountain tundra. On the other hand, the litter mass in the tundra accounted for 98% of the mortmass, and decreased to 46% in the valley. Plant biomass prevails over mortmass at all forest levels of the transect (factor 3.33–2.67), but mortmass reaches the highest ratio at the mountain-tundra level (factor 3.81). The excess of mortmass over biomass is also shown in some high-altitude communities of the Altay-Sayan mountains [36,37], the tundra communities of Fennoscandia [38], and the zonal tundra habitats of the Yamal peninsula [39].

### 3.3. Fungi

During research in 2020, 18 new species of aphyllophoroid fungi were identified for the study areas: *Athelia bombacina* (Link) Pers., *Botryobasidium botryosum* (Bres.) J. Erikss., *B. laeve* (J. Erikss.) Parmasto, *Clavicorona taxophila* (Thom) Doty, *Datronia mollis* (Sommerf.) Donk, *Gloeocystidiellum leucoxanthum* (Bres.) Boidin, *Hydnum rufescens* Pers., *Hyphodontia arguta* (Fr.) J. Erikss., *H. breviseta* (P. Karst.) J. Erikss., *Kavinia alboviridis* (Morgan) Gilb. & Budington, *Leptosporomyces fuscostratus* (Burt) Hjortstam, *Phanerochaete laevis* (Fr.) J. Erikss. & Ryvarden, *Phlebia nitidula* (P. Karst.) Ryvarden, *Phlebiella borealis* K.H. Larss. and Hjortstam, *Polyporus choseniae* (Vassilkov) Parmasto, *Thelephora caryophyllea* (Schaeff.) Pers, *Tomentella stuposa* (Link) Stalpers, *Tomentellopsis echinospora* (Ellis) Hjortstam (Appendix A).

Taking into account the whole 60-year investigation period at the Slantsevaya mountain, a total of 281 aphyllophoroid fungal species were identified in an area of only 10 km^2^ [9]. Together with fungi collected in areas with a anthropogenic impact only [40,41,42], the total number is 294 species. Thus, this site is certainly one of the best studied northern areas in Russia concerning aphyllophoroid fungi. For comparison, 400 species are known from the Murmansk province (north of the Arctic Circle) on an area of 145,000 km^2^ [43], and 550 species were reported from mid-boreal forests of the Republic of Karelia on 172,400 km^2^ [44].

On the investigated altitudinal gradient, 157 species were identified (Appendix A). The number of fungal species is significantly negatively correlated with altitude (*r*_s_ = −0.9, *p* = 0.037): from the river valley towards the tundra, the number of aphyllophoroid fungal species decreases from 106 to 30 (Table 4). The same general dynamics was also observed for the morphological forms of aphyllophoroid fungi (Figure 5A). However, morphological forms showed different dynamics depending on altitudinal level (Figure 5B). Thus, the proportion of clavarioid fungi is maximally positively associated with altitude (*r*_s_ = 0.97, *p* = 0.0048), and that of poroids is strongly negatively correlated to it (*r*_s_ = −1.0, *p* < 0.001). No correlation was found for corticoid fungi (*r*_s_ = 0.21, *p* = 0.74), although this parameter decreases within the forest zone towards the lowlands (from 49.1% to 36.7%), and is strongly correlated with altitude (*r*_s_ = 1.0, *p* < 0.001).

The diversity of individual morphological forms of aphyllophoroid fungi changes in a clear correlation with altitude, even though the most species-rich altitudinal level is level II and not I.

### 3.4. Biotic Factors Influencing Aphyllophoroid Species Richness

#### 3.4.1. Plant Diversity

Aphyllophoroid fungal species richness is not correlated to plant diversity (*r*_s_ = −0.05, *p* = 0.93) (Figure 6A). However, a positive correlation was found between aphyllophoroid fungal species richness and richness of tree species (*r*_s_ = 0.95, *p* = 0.013), but there was no relationship to forbes and grass species richness (*p* = 0.39 and 0.56, respectively). Moreover, a reliable relationship was detected between aphyllophoroid fungal species richness and therophytes (*r*_s_ = 0.89, *p* = 0.04), and poroid fungal species were a significantly correlated with mesophanerophyte (*r*_s_ = 0.83, *p* = 0.02) and tree diversity (macrophanerophyes) (*r*_s_ = 0.95, *p* = 0.014). Moreover, clavarioid fungal species richness was significantly related to tree species richness (macrophanerophyes) (*r*_s_ = 0.84, *p* = 0.005), while no such correlation was found for corticioids. With an increase in the proportion of phanerophytes, the proportion of clavarioids decreases, which is similarly shown for the Urals in the natural zones of the subzones [10]. The number of clavarioids is related to the number of geophyte species (*r*_s_ = 0.84, *p* = 0.005). The number of tree species is most strongly related to the number of parasitic and wood-destroying fungi (*p* = 0.01), and with the sum of all saprobes (*r*_s_ = 0.81, *p* = 0.05). Moreover, the number of species of height group 3 (high-size fruitbodies) is positively associated with the number of tree species (*p* = 0.03).

#### 3.4.2. Altitudinal Levels

The richest aphyllophoroid mycobiota occurred in levels I and II, with highest crown density indices (*r*_s_ = 0.9–0.95, *p* = 0.002), and the highest values of CWD and FWD. In contrast, thereto has tundra with the poorest aphyllophoroid fungal communities, which appeared to be related to the low crown density, minimal mortmass of CWD and FW and the highest amounts of lichen, moss and litter biomasses (Table 3).

Altitudinal levels had a large impact on fungal community structure: the most antagonistic characteristics were found between the altitudinal levels V (tundra) and II (the closest forest stand) (Figure 7), where distinct differences in fungal morphological forms were detected. The poorest fungal communities were found in the altitudinal level V with richest lichen diversity. Axis 1 in the ordination represents stock characteristics and the stand density of forest communities. Level II, located in the middle part of the mountain slope, is the oldest north-boreal coniferous forest with the highest tree crown densities (80–90%) and high amounts of woody biomass (>130 t/ha). The aphyllophoroid species richness of altitudinal level II was shaped by a maximum species richness of grasses and bushes, and the highest amounts of CWD and FWD.

The aphyllophoroid fungal species richness in altitudinal level V, in which litter saprobs are clearly dominating, can be explained by the high litter mortmass and the high biomass of lichens and bryophytes (Figure 8). In the tundra, biomass is generally low and consists of thin twigs and leaves and individual blades of grass. This resulted in a prevalence of small-size fruitbodies.

#### 3.4.3. Plant Biomass

Due to the heterotrophic nutritional mode of aphyllophoroid fungi, it would be obvious to assume a close relationship between the species richness of saprobic fungi and the amount of mortmass. However, such a relationship was not found (*r*_s_ = −0.7, *p* = 0.188) (Figure 6B), nor was mortmass related to the total number of aphyllophoroid fungi (*r*_s_ = −0.68, *p* = 0.236).

A relationship was detected only between for certain trophic (substrate) groups of fungi and certain mortmass fractions. For example, the number of corticoid, poroid and clavarioid fungi was closely related to CWD and FWD (*p* < 0.05), while clavarioids were strongly related to litter mortmass (*r*_s_ = 0.97, *p* = 0.004), showing the strongest positive correlation (*r*_s_ = 1, *p* = 0) when considering proportions (proportion of litter to the total mortmass and proportion of clavarioid species to the total number of aphyllophoroid fungi).

Aphyllophoroid fungal species richness was clearly related to biomass (*r*_s_ = 1, *p* < 0.001), but no relationship was found between the number of fungal species and the total aboveground plant biomass (*r*_s_ = 0.6, *p* = 0.284). A significant positive relationship exists to total aboveground biomass (*r*_s_ = 0.9, *p* = 0.037).

Wood-decaying fungi were significantly correlated to the amounts of CWD and FWD (*r*_s_ = 1.00, *p* < 0.001), while litter saprobes were strongly correlated to litter mortmass (*r*_s_ = 0.9, *p* = 0.037). The proportion of wood-decaying saprobes was stable at all four altitudinal levels with forests (45.4–46.6%).

The biomass of living trees was also positively correlated to the richness of fungal species with large-size fruitbodies (*p* < 0.001). Due to the higher biomass, there were significantly more fungal species of the III height group at the bottom of the mountain (F (4.13) = 5.63; *p* = 0.01).

The difference between the aphyllophoroid communities occurring in the treeless tundra and in the boreal zone can be summarized as follows. In the poorest level, the tundra, litter-inhabiting clavarioids with small-size basidiomata (I height group), are highly represented (48% of the total species), while other ecological and morphological groups of fungi do not exceed 7% of the total species. In contrast, in the richest boreal level II, the ecological and morphological groups of aphyllophoroids are more evenly distributed: wood-inhabiting corticioids and poroids (31% and 25%, respectively), and litter-inhabiting clavarioids with middle-size basidiomata (II height groups) (19%). Saprobes are dominating at all altitudes, while, as expected, the richness of mutualistic symbionts or parasites is generally low in this subarctic study site.

## 4. Discussion

This study proved the hypothesis that aphyllophoroid fungal species richness is generally decreasing with altitude, as is also reported from other parts of the Urals [9]. However, altitude is not always related to the number of fungal species, as this also depends on the scale and structure of the investigated area. In an earlier study carried out in an adjacent mountain of the Polar Urals, we examined forest areas (20 × 20 m) with crown densities of 70–80%, and a CWD volume of about 100 t/ha, and compared them to several areas in the middle and lower parts of the slope, where open meadows and wetlands were predominating, and CWD was 20 t/ha. Here, species richness was significantly lower in lowlands compared with the upper part of the forest belt [10]. Similar results have been shown in other studies [45]. Thus, aphyllophoroid fungal species richness is also related to the structure of the forest, and to the amount of CWD.

Moreover, the other hypotheses we tested were largely confirmed. As hypothesized, fungal species richness negatively correlates with altitude. The highest species richness was detected in old-growth forests with a maximum crown density. A direct relationship between plant richness and aphyllophoroid richness was not found. However, the study confirmed close correlations for individual ecological groups, e.g., for tree species richness and wood-destroying fungi, or for geophytes and clavarioid fungi. We also proved that both the richness of wood-decaying fungi, and the richness of fungi-forming large fruitbodies, correlates to wood volume.

### 4.1. Why Is Plant Diversity Highest at the Slopes of the Mountains?

The richness of aphyllophoroid fungi was highest at lower altitudinal levels, and was gradually decreasing from the slopes to the tundra. At first glance it appears strange that species diversity is lower in the river valley than at the slopes. However, this can be explained by the typical structure of this pristine river valley, as the ground is largely covered with mosses and horsetail. Over the past 15 years, its swampiness has increased due to an increase in precipitation, increase permafrost thawing depth and increasing amounts of water running down the slopes into the valley. This explains why optimal conditions for the aphyllophoroid communities are rather found on the middle slope of the mountain. Here, the conditions are optimal due to an ideal SW exposure, drainage and no stagnation of cold valley air, and a maximal plant richness (especially of trees, shrubs, grasses and forbs). On the slopes, soil permafrost has been thawing fastest over the last 35 years, thus resulting in high soil microbial activities [9]. This is reflected by the maximum number of aphyllophoroid species growing on soil in these areas (humus saprobic and ectomycorrhizal species like *Ramaria, Hydnum, Thelephora* spp.), and by the recent “invasion” of “southern forest” species (*Polyporus umbellatus* (Pers.) Fr., *Osteina obducta* (Berk.) Donk, *Coltricia perennis* (L.) Murrill).

### 4.2. Is Fungal Diversity Positively Correlated to Plant Diversity?

Richness of tree species appears to be a key factor for aphyllophoroid fungal species richness. The number of tree species is strongly related to the richness of poroid fungi, to the richness of plant parasitic fungi and to richness of corticoid fungi. This is probably all because of the strong dependence of these fungi on plant biomass. Due to the higher biomass, there are significantly more fungi with large fruitbodies at the bottom slopes of the mountain. A similar trend was shown for clavarioid fungi along the entire Ural latitudinal-zonal transect [46]: the proportion of clavarioid species of the III height group increases to a maximum in the direction from the tundra to hemiboreal forests, while clavarioid species of the I height group decreases to a minimum. As shown on the Slantsevaya mountain, negatively geotropic fungi, like stipitate *Hydnum, Thelephora, Cantharellus, Polyporus*, etc., are also following this trend.

Species richness of saprobial fungi is not related to the amount of mortmass. There are several hypotheses which might explain this. We assume contrasting trends in different altitudinal levels to be the main reason for this paradox: in the lower part of the slope with CWD and FWD as predominating mortmass, this relationship is clearly positive; however, higher up the slope, where litter, grass, lichens and mosses predominate mortmass, this relationship is negative. Further research is needed to prove this hypothesis.

### 4.3. The subarctic as a Model for Biotic Factors Influencing Aphyllophoroid Fungal Communities

The tree species dominating in the studied subarctic region is Siberian larch, and the main undergrowth is mountain green alder [2]. Decomposition rates of *Larix* wood are important ecological factors for ecosystem development and changes in subarctic regions [47]. The study area almost represents the western border of the “Siberian” range of these two tree species. West of the Urals, the green alder bushes disappear from the undergrowth, and also Siberian larch sharply reduces its coenotic role; its westernmost locations are in the White Sea region. A modeling-based prediction of future changes in similar areas is extremely interesting and could also be economically important.

In the upper part of the closed forest, the basis of the biomass is formed by larch (and green alder), but as soon as spruce and large birch appear, the number of wood-destroying fungal species increases and reaches a maximum in areas with the highest species richness of trees and bushes and with the maximal CWD and FWD. Thanks to the appearance of spruce, CWD increased by only 7.6 t/ha (6.6% of the total woody biomass formed by three tree species, see Appendix A), but this was enough for a significant increase in the species richness of wood-destroying fungi. Thus, the number of tree species turned out to be an important predictor for the species richness of poroid fungi growing on rotting logs, as well as for their share of the total number of species. In Siberian subarctic forests, fungal species richness generally strongly depends on the forest type (conifers vs. deciduous trees), and on the presence of single tree species [48,49]. The identity of tree species can even be the main factor shaping the composition of wood-decaying fungi [11]. The overall amount and diversity of deadwood (including the role of different tree species) is a major factor shaping the diversity of wood-decaying fungi [50,51,52,53,54,55,56]. In addition, the size of the wood explains the diversity of fungi belonging to different morphological or functional groups: as is convincingly shown in this study, species richness of poroid fungi significantly depends on log size.

Despite the small size of the studied area, striking similarities were found with results from similar large-scale studies [10]. On the 3000 km long Ural latitudinal transect, the maximum proportion of poroid fungi (of the total number of aphyllophoroid species) corresponds to southern and hemiboreal forests with the maximum crown density, stand age and species diversity of tree and shrub species, while in open tundra regions their proportion is minimal. The maximum proportion of clavarioid fungi was found in the “treeless” tundra regions, and it slowly decreases from the forest tundra to hemiboreal regions. The proportion of corticoid fungi was maximal in the forest-tundra and was slowly decreasing towards the hemiboreal forests. This confirms that the extensive data obtained in this study in a small area are correct, and that conclusions are also valid for larger scales. This study revealed and confirmed important factors shaping the mycobiota in subarctic regions, and is thus forming a solid base for future large-scale studies and simulations.

## 5. Conclusions

Aphyllophoroid fungi represent all three main ecological groups of macromycetes, allowing them to develop on the maximum possible spectrum of substrates available. In this regard, they can be considered as a simple ecological model describing the principle fungal distribution in a simply arranged, high-latitude forest ecosystem, at the polar limit of the taiga.

Strikingly, we discovered a ratio of 1:3–3.5 between the aphyllophoroid fungal species richness of the tundra and the adjacent river-valley. The middle part of the slope appears to have optimal conditions for these fungi, as the maximum species richness occurred here. However, fungal species richness was not related to the species richness of plants or to mortmass in these habitats. The poorest aphyllophoroid communities were related to the richest lichen communities in terms of lichens species richness and biomass, as well as to evergreen ericoids and *Sphagnum* bogs.

The range of morphological and ecological groups of aphyllophoroid fungi occurring in the tundra differs significantly from those occurring in forest. This shows that fungal communities are highly influenced by various biotic factors such as substrate availability and substrate type. Most of the detected fungi were wood saprobes, but also litter saprobial species were quite species-rich, while humus saprobes, parasites and ectomycorrhizal aphyllophoroid fungi were comparatively rare.

At the arctic limit of forest distribution, the species richness of aphyllophoroid fungal communities is determined by the amount of available dead wood, and with the growth of this resource, the proportion (and absolute species richness) of poroid fungi increases. At the same time, the proportion of clavarioid fungal species is higher in the tundra and decreases with an increase in the density of the forest crowns. The higher the litter mortmass, the higher the proportion of clavarioid fungi and their species richness in the aphyllophoroid communities becomes. The proportion of corticoid fungal species reaches a maximum in crooked forests, and was negatively correlated to the biomass of woody plants. However, the correlation of the species richness of aphyllophoroid fungi with the floristic richness and total mortmass could not be established. This is an interesting result for aphyllophoroid macromycetes, as their species richness and diversity were, up to now, traditionally associated with the availability of substrate resources only.

## Figures and Tables

**Figure 1 jof-06-00353-f001:**
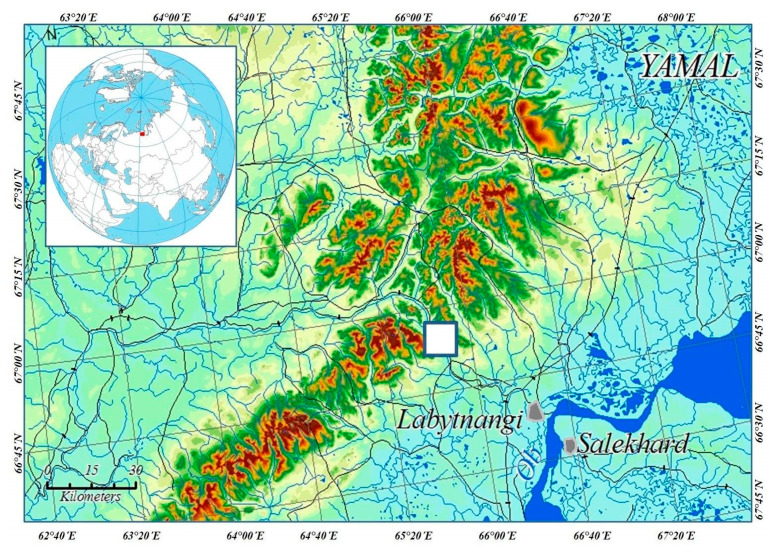
Map of the investigated area in the Polar Urals. The study area is marked as a white square.

**Figure 2 jof-06-00353-f002:**
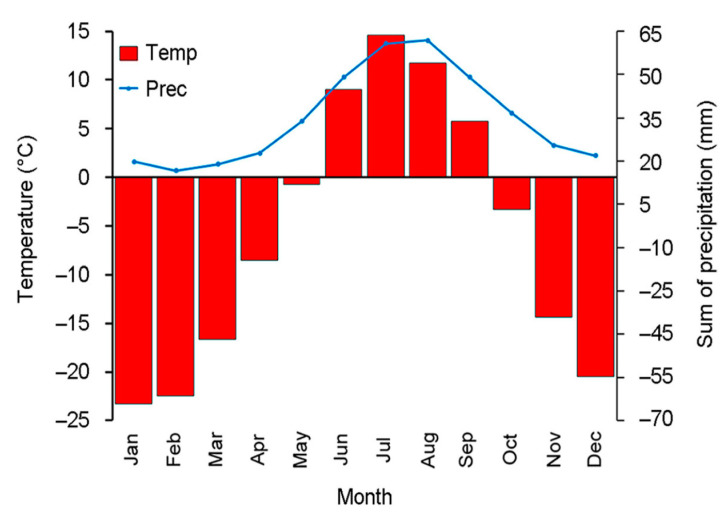
Climagram for Salekhard weather station (1892–2018). The blue line represents the long-term average sum of precipitation, red columns are the average monthly temperatures.

**Figure 3 jof-06-00353-f003:**
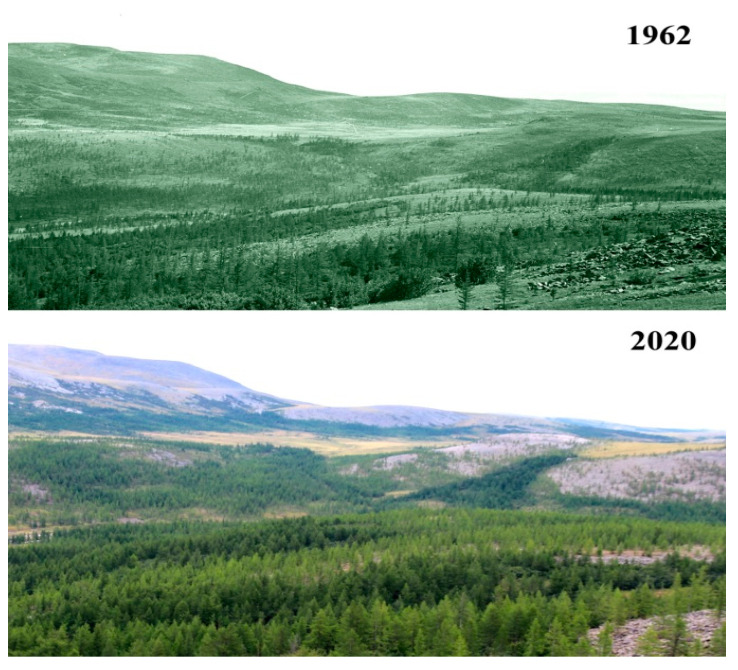
Comparison of the development of vegetation on the Slantsevaya mountain over 58 years (1962–2020) based on photographs. The rise of the upper forest border, and an increase in the tree crown density are evident.

**Figure 4 jof-06-00353-f004:**
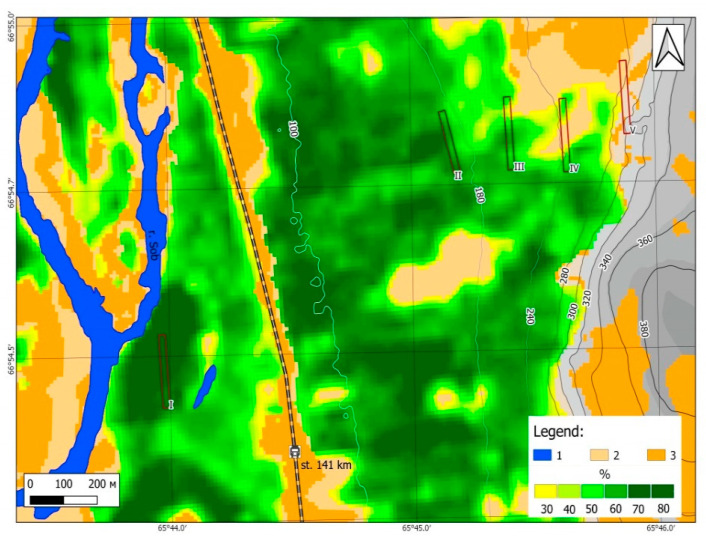
Map of Slantsevaya mountain with the studied altitudinal levels as well as parameters of forest crown density. Color legend: blue 1—water (Sob’ river), beige 2—grass communities, orange 3—stones; yellow to dark green transitional colors represent tree crown density (%).

**Figure 5 jof-06-00353-f005:**
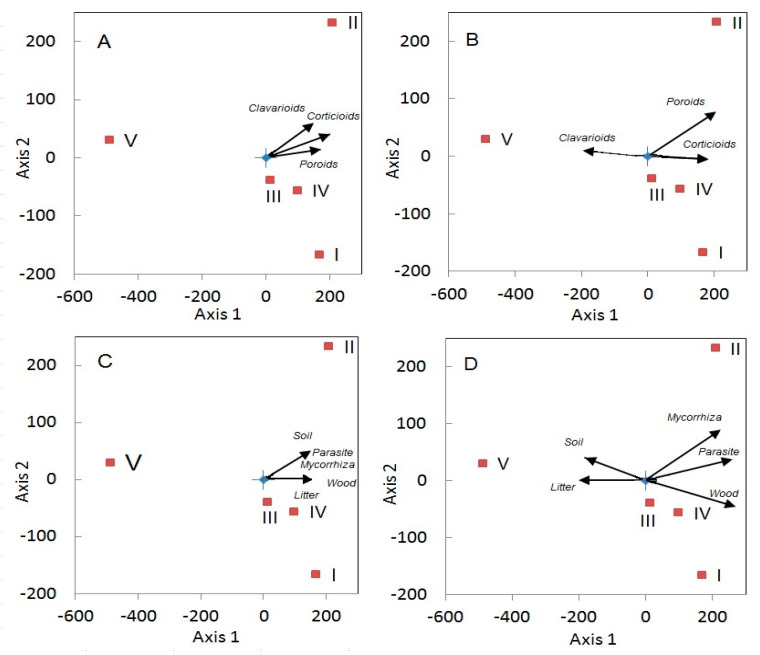
Relationship between aphyllophoroid fungal species richness at different altitudinal levels (I–V) and the structure of the mycobiota. (**A**) Species richness of fungal morphological groups (corticoid, poroid, clavarioid); (**B**) Proportion of fungal morphological groups; (**C**) Species richness of fungal ecological groups (wood, litter or soil saprobes, parasites, mycorrhizal); (**D**) Proportion of fungal ecological groups.

**Figure 6 jof-06-00353-f006:**
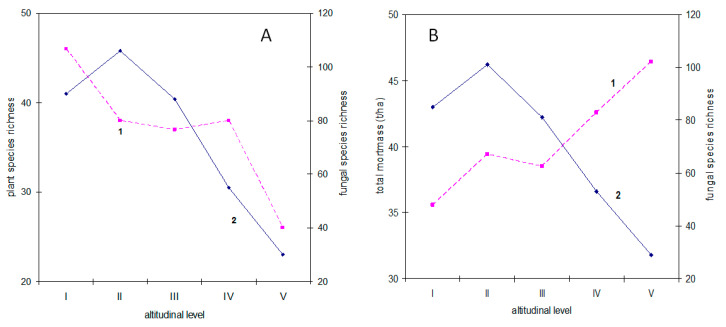
Vascular plant richness, fungal richness and total mortmass at different altitudinal gradients I–V. (**A**) 1—Vascular plant species richness; 2—Fungal species richness; (**B**) 1—Total mortmass; 2—Species richness of saprobic aphyllophoroid fungi.

**Figure 7 jof-06-00353-f007:**
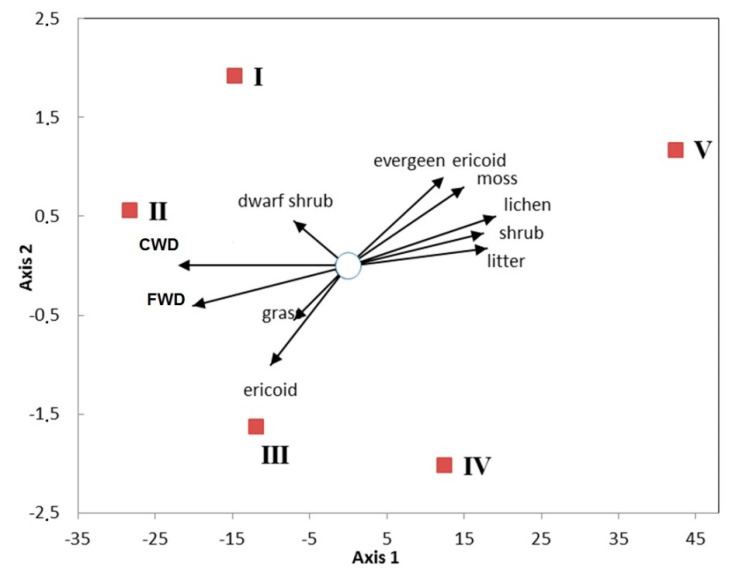
Relationship of the richness of morphological groups of fungi with biomass and mortmass of vegetation. Axis 1—biomass of the tree layer, Axis 2—morphological groups of fungi. I–V represent the altitudinal levels.

**Figure 8 jof-06-00353-f008:**
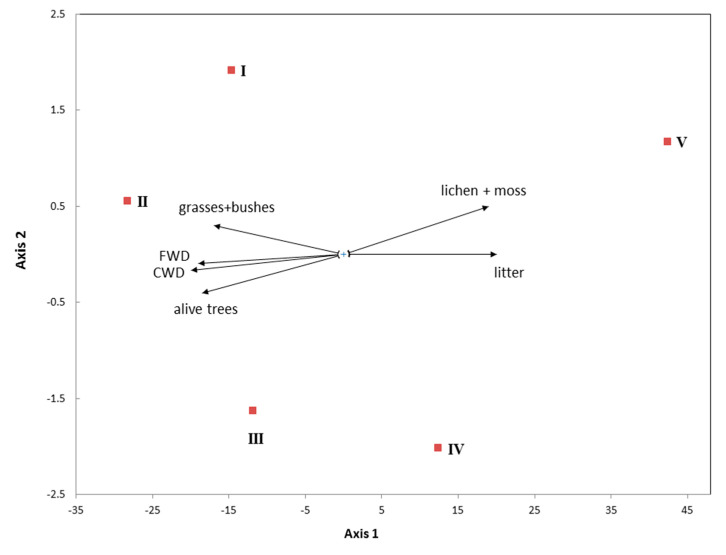
Relationship between different ecological groups of fungi or groups with substrate preference with plant biomass and mortmass in the altitudinal levels (I–V).

**Table 1 jof-06-00353-t001:** Number of vascular plant species detected at the different altitudinal levels of the Slantsevaya mountain. Data are presented as richness of plant functional groups, bryophyte and lichen richness and plant life forms.

	Altitudinal Level				
	I	II	III	IV	V
Altitude (m a.s.l.)	80	170	230	260	310
Vascular plants species richness	46	38	37	38	26
	Functional groups				
Tall shrubs	2	5	3	4	1
Shrubs	8	4	6	3	5
Dwarf shrubs	1	1	1	1	1
Evergeen ericoids	1	2	3	1	7
Forbs	28	23	19	26	5
Grasses	5	2	4	2	6
Ericoids	1	1	1	1	1
	Bryophytes and lichens				
Bryophytes	8	7	4	4	7
Lichens	0	0	0	0	21
	Life forms				
(meso + nano) Phanerophytes	9	8	8	6	6
Hemicryptophytes	21	15	15	20	9
Chamaephytes	7	9	8	7	9
Geophytes	5	2	3	2	2
Therophytes	2	3	2	2	0
Cryptophytes	1	1	1	1	0

Note: altitudinal level: I—river valley, II—middle part of the slope, III—upper limit of closed forest, IV—mountain crooked forest, V—mountain shrub tundra.

**Table 2 jof-06-00353-t002:** Vascular plant aboveground biomass (g/m^2^) of different functional groups and life forms.

	Altitudinal Level				
	I	II	III	IV	V
	Functional group				
Tall shrubs + Shrubs	7.8 ± 0	2.2 ± 0.8	9 ± 3.8	5.7 ± 4	13.8 ± 4.6
Dwarf shrubs	6.4 ± 4.8	144.4 ± 0	0.2 ± 0	0	44.6 ± 41.7
Evergreen ericoids	39.2 ± 10.4	5 ± 3.8	1.2 ± 1	0	70.2 ± 14.9
Ericoids	9.6 ± 0	28.2 ± 2.5	26.2 ± 0	36.2 ± 33.4	0
Grasses + Forbs	75.1 ± 16.1	425.4 ± 230.6	134.9 ± 49.2	165.5 ± 38.8	7.7 ± 1.6
	Life form				
(meso + nano) Phanerophytes	7.6 ± 3.6	0	57.6 ± 47.2	0	49.7 ± 17.6
Hemicryptophytes	29.7 ± 7.2	140 ± 106.5	81.9 ± 17.3	102.9 ± 38	6.6 ± 1.5
Chamaephytes	38.4 ± 17.8	348.9 ± 163.8	70.1 ± 52.7	81 ± 20.9	103.6 ± 44
Geophytes	60.4 ± 13.4	25.2 ± 22.6	71.2 ± 32.6	31 ± 19.7	14.8 ± 14
Therophytes	12.8 ± 6.7	7.8 ± 5.4	13 ± 2.2	3.6 ± 0	0
Cryptophytes	0.1 ± 0	0	0	0.7 ± 0.3	3.2 ± 0

Note: altitudinal level: I—river valley, II—middle part of the slope, III—upper limit of closed forest, IV—mountain crooked forest, V—mountain shrub tundra.

**Table 3 jof-06-00353-t003:** Distribution of aboveground plant mortmass, biomass and total phytomass (t/ha) along the altitudinal gradient.

	Altitudinal Level				
Parameter	I	II	III	IV	V
	Plant mortmass				
CWD	9.6	10.7	8.5	7.1	0.2
FWD	9.5	8.7	7.5	6.6	0.7
Litter	16.5	20.0	22.5	28.9	45.5
Sum:	35.6	39.4	38.5	42.6	46.4
	Plant biomass				
Alive trees	107.7	114.6	102.25	100.67	1.73
Grasses + bushes	9.33	15.32	7.11	6.54	5.23
Lichens + mosses	1.62	0.31	0.56	0.35	6.56
Sum:	118.65	130.33	116.48	113.75	12.19

Ratio plant biomass/mortmass	3.33	3.31	3.02	2.67	−3.81
	Total aboveground phytomass				
Sum of plant biomass and mortmass	154.25	169.73	154.98	156.35	58.59

Note: altitudinal level: I—river valley, II—middle part of the slope, III—upper limit of closed forest, IV—mountain crooked forest, V—mountain shrub tundra. CWD—coarse woody debris; FWD—fine woody debris.

**Table 4 jof-06-00353-t004:** Species richness and proportion (%) of aphyllophoroid fungi belonging to different morphological forms, to different ecological strategies, having different substrate preferences or with different fruitbody size groups.

	Altitudinal Level				
	I	II	III	IV	V
Altitude (m a.s.l.)	80	170	230	260	310
Species richness	90	106	84	55	30
	Morphological form				
Corticioids	33/36.7	42/39.6	35/41.7	27/49.1	11/36.7
Poroids	29/32.2	29/27.4	22/26.2	8/14.5	1/3.3
Clavarioids	28/31.1	35/32.1	27/32.1	20/36.4	18/60.0
	Ecological strategy and substrate groups				
Saprobs	110/87.4	131/88.4	107/90.7	70/90.9	29/91.9
Wood	56/45.5	68/45.9	55/46.6	35/45.4	8/23.5
Litter	47/36.2	53/35.8	44/37.3	31/40.3	18/53.5
Soil	7/5.7	10/6.7	8/6.8	4/5.2	3/9.1
Parasites	5/3.9	5/3.5	3/2.5	2/2.6	1/2.6
Symbionts	11/8.7	12/8.1	8/6.8	5/6.5	3/5.5
	Fruitbody size				
I	16/48.0	19/51.3	17/62.9	14/70.0	16/88.8
II	13/40.4	15/40.6	10/37.1	6/30.0	2/11.2
III	4/11.6	3/8.1	0	0	0

Note: altitudinal level: I—river valley, II—middle part of the slope, III—upper limit of closed forest, IV—mountain crooked forest, V—mountain shrub tundra.

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
