# Peer review of "Relationship between Species Richness, Biomass and Structure of Vegetation and Mycobiota along an Altitudinal Transect in the Polar Urals"

_jof, 2020, doi:10.3390/jof6040353_

Round 1

Reviewer 1 Report

The study by Shiryaev and colleagues reports on the relationship between species richness, biomass and structure of vegetation and mycobiota along an altitudinal transect in the Polar Urals. In particular, Authors concentrated their sampling and analysis efforts on aphyllophoroid macrofungi, studying their richness (species number) at five levels along an altitudinal gradient (80-310 m a.s.l.) and correlating it to the species richness patterns of vascular plants, bryophytes and lichens.

Overall, the results are interesting, disclosing several non-granted patterns and adding significantly to our knowledge of ecosystems’ structure and dynamics in an area particularly subjected to the effects of climate change.

Some quibbles hover in the mind of this reviewer.

- Do the Authors have any data on the abundance (in terms of number of sporocarps or total biomass) of aphyllophoroid fungi in the plots they studied? Could this analysis modify some of the conclusions they drew?

- Why only aphyllophoroid macrofungi were considered? What about species-rich genera of mycorrhizal macrofungi (e.g. Cortinarius, Lactarius, Russula, et cetera)? Where these present at the study place? I guess that if the answer is ‘yes’, taking these into count could have significantly affected the analysis in terms of fungi ecological guilds.

- Authors should cite references for their attribution of selected fungal species to ecological roles. For example, on the basis of which literature some Ramaria species where reported as mycorrhizal and others as litter-inhabiting (see Supplementary material; to note that sheets are numbered in Cyrillic). Or, who says that Tomentella is both mycorrhizal and wood-decaying?

Author Response

Answers to Reviewer #1

- Do the Authors have any data on the abundance (in terms of number of sporocarps or total biomass) of aphyllophoroid fungi in the plots they studied? Could this analysis modify some of the conclusions they drew?

Answer: Yes, we have data on the abundance, but at the moment, a lot of data is already included in the manuscript. We decided to write a separate article on the abundance dynamics of fungi along this high-altitude transect. Otherwise, this article would be overloaded with data. At the moment, we continue to process the data of fungal abundance, but according to the primary results, it can be stated that many of the results presented in this manuscript will only increase with the addition of abundance parameters. For example, the number of “forest” soil- and wood-destroying species is maximal in the middle part of the slope, while the abundance of arcto-alpine species, clavarioids litter-inhabiting, is maximal above the forest boundary.

- Why only aphyllophoroid macrofungi were considered? What about species-rich genera of mycorrhizal macrofungi (e.g. Cortinarius, Lactarius, Russula, et cetera)? Where these present at the study place? I guess that if the answer is ‘yes’, taking these into count could have significantly affected the analysis in terms of fungi ecological guilds.

Answer: In our study, aphyllophorpoid fungi are a model group. It has been most well studied in 60 years on the mountain slopes and in the river valley. We know for sure the long-term dynamics of the species richness of this group, therefore, we can reasonably discuss about the role of climate change or the effect of phytomass on their species richness. The purpose of this work is to establish the first result, to reveal the relationship between the fungal diversity and biotic factors. We did it. Next, we will replenish the list of species with agaricoids and gasteroids fungi. This is already being done, therefore, the results for 3 groups of basidial macromycetes will be presented in the near future.

- Authors should cite references for their attribution of selected fungal species to ecological roles. For example, on the basis of which literature some Ramaria species where reported as mycorrhizal and others as litter-inhabiting (see Supplementary material; to note that sheets are numbered in Cyrillic). Or, who says that Tomentella is both mycorrhizal and wood-decaying?

Thank you, we included this information in the text.

Reviewer 2 Report

It is very interesting result as the species richness and diversity were, up to now,
traditionally associated with the availability of substrate resources. 

I happy to recommend getting this published with the current format. 

Author Response

Answers to Reviewer #1

Thanks for your positive comments.